# Benzoic Acid Combined with Essential Oils Can Be an Alternative to the Use of Antibiotic Growth Promoters for Piglets Challenged with *E. coli* F4

**DOI:** 10.3390/ani10111978

**Published:** 2020-10-28

**Authors:** Leticia Mendonça Rodrigues, Túlio Otávio de Araújo Lima Neto, Cesar Augusto Pospissil Garbossa, Claudia Cassimira da Silva Martins, Dino Garcez, Laya Kannan Silva Alves, Márvio Lobão Teixeira de Abreu, Rony Antonio Ferreira, Vinícius de Souza Cantarelli

**Affiliations:** 1Department of Animal Science, Federal University of Lavras, Lavras 37200-000, Brazil; leticiamendoncarodrigues@gmail.com (L.M.R.); tulio.neto16@hotmail.com (T.O.d.A.L.N.); marvio@ufla.br (M.L.T.d.A.); rony@ufla.br (R.A.F.); 2Department of Animal Nutrition and Production, School of Veterinary Medicine and Animal Science, University of São Paulo, Pirassununga 13635-900, Brazil; cgarbossa@usp.br (C.A.P.G.); layakannan@usp.br (L.K.S.A.); 3DSM Nutritional Products, São Paulo 01452-001, Brazil; claudia.silva@dsm.com (C.C.d.S.M.); dino.garcez@hotmail.com (D.G.)

**Keywords:** bacteria, microbiology, organic acid, pig nutrition, stress physiology, weaning

## Abstract

**Simple Summary:**

The use of antibiotics as growth promoters for swine must be minimized as it can promote resistance in microorganisms. Therefore, it is essential to search for alternative additives. This study aimed to investigate the effects of benzoic acid and a blend of essential oils (thymol, 2-methoxyphenol, eugenol, piperine, and curcumin) on the performance and intestinal health of weanling piglets challenged with *Escherichia coli* F4. The combination of benzoic acid and essential oils at 3 g/kg improved the piglets’ average daily gain and consequently their final body weight and it is an economically viable alternative to replace colistin. These results could have a great impact on society, contributing to the one heath concept and demonstrating the ability to replace antibiotics as growth promoters and thus minimize the chance of causing bacteria resistance.

**Abstract:**

Benzoic acid (BA) and essential oils (EOs) (thymol, 2-methoxyphenol, eugenol, piperine, and curcumin) are being studied to minimize the impairment of gastrointestinal functions in weanling piglets. This study evaluates the effects of combining BA and EO on the performance and intestinal health of piglets challenged with *E. coli* F4 (1 mL, 10^6^ CFU/mL). In total, 270 weaned piglets were used in a randomized block design with six treatments: positive control, with 40 mg/kg colistin (PC); negative control, without the growth promoter (NC); negative control +5 g/kg benzoic acid (BA); negative control +2 g/kg combination of BA+EO (BA+EO2); negative control +3 g/kg combination of BA+EO (BA+EO3); negative control +4 g/kg combination of BA+EO (BA+EO4). BA+EO3 presented a greater average daily gain (ADG) (*p* = 0.0013) and better feed-to-gain ratio (*p* = 0.0138), compared to NC, from 21 to 35 days age. For the total period, BA, BA+EO3, and BA+EO4 were similar to PC and superior to NC for ADG (*p* = 0.0002) and final body weight (BW) (*p* = 0.0002). No difference (*p* > 0.05) was verified for diarrhea, microbial population, production of volatile fatty acids, pH, weight of organs, cellular proliferation, and cholecystokinin count. NC and BA+EO4 resulted in a higher villus height in the jejunum (*p* = 0.0120) compared to BA+EO3. The use of BA or the combination of BA and EO at 3 g/kg provides improved performance, aside from being an economically viable alternative to replace colistin.

## 1. Introduction

The gastrointestinal tract (GIT) is a complex environment, especially after weaning, when there is separation of the piglet from the sow and the rest of the litter, a change of facility, the imposition of a new social life, and an abrupt change from a liquid diet to one based on less digestible plants [1]. With the occurrence of these stressful events, there is a fall in feed intake in the first two nursery weeks and some physiological functions of the GIT are not remodeled quickly enough to keep up with the changes and to maintain the piglet’s performance [2]. Villi atrophy, deterioration of barrier function, and electrolyte absorption and secretion disturbances are triggered [3], leading to an increased susceptibility to infectious agents and generating clinical signs such as diarrhea and reduced feed intake [4,5].

The occurrence of diarrhea is a serious consequence, which can lead to morbidity and mortality in piglets. Most occurrences in this phase are caused by enterotoxigenic *Escherichia coli*, a highly proliferating pathogenic bacterial strain that has fimbriae for adhering to the intestinal mucosa and proliferating [6]. For many years, the administration of non-therapeutic concentrations of antimicrobial agents, known as growth-promoting antibiotics (GPAs), has been and remains the main line of defense in swine farming [7,8].

Colistin has been a widely used antibiotic in agriculture since 1950, performing specific actions against Gram-negative bacteria (GNB) such as *Enterobacter aerogenes*, *Escherichia coli*, and species of Salmonella [9,10,11]. The mechanism involves passage through the lipopolysaccharide barrier of the outer membrane, followed by denaturation of the plasma membrane, extravasation of cellular constituents, and cell death [9,12].

According to Kempf et al. [13], the overuse of colistin in both human and veterinary medicine has quickly caused the development of resistance in GNB to current antibiotics. Given the importance of this antibiotic, it has been reclassified onto the List of Critically Important Antimicrobials for Human Medicine by the World Health Organization [14].

The emergence of antibiotic-resistant bacterial strains has led to many important pork markets banning colistin as a growth promoter. Liu et al. [15] described bacterial resistance to colistin in animals and humans, highlighting the resistant plasmid among strains of *Escherichia coli*. To minimize its use on piglets and economic problems in this stage, as well as to seek alternatives to antibiotics, benzoic acid and essential oils have been used. Studies show that benzoic acid has an antibacterial action and positive impacts on the performance of weanling piglets, showing a 12% improvement in average daily gain (ADG) [16,17,18,19]. Microbiota modulation and improved performance (11% greater ADG) have also been observed in poultry and pigs supplemented with essential oils [20,21,22,23].

Due to the lipophilic characteristic of essential oils and benzoic acid when not dissociated, both can diffuse through the bacterial cell wall, altering its permeability. The release of H^+^ ions inside the cell makes the bacteria attempt to resume homeostasis, removing protons through the Na^+^/K^+^ pump. An active process that promotes bacterial depletion diminishes its pathogenicity and compromises vital processes [21,24,25], characterizing a bactericidal or bacteriostatic effect. Thus, the association of benzoic acid and essential oils shows synergistic activity in which essential oil compounds can damage and make the pathogenic organisms’ cell walls more permeable, allowing easier entry and dissociation of the acid inside the cell [26,27].

This study aimed to investigate the effects of combining benzoic acid and a blend of essential oils, at different concentrations, compared to benzoic acid alone at a customary dose and to an antibiotic. Based on an evaluation of the treatment effects on the performance and intestinal health of weanling piglets challenged with *Escherichia coli* F4, the study strives to understand if the dose of benzoic acid can be reduced when associated with essential oils.

## 2. Materials and Methods

### 2.1. Animals and Feed

The experiment was conducted in the Swine Experimental Center at the Animal Science Department of the Federal University of Lavras (UFLA, Lavras, Brazil), in Lavras, Minas Gerais, Brazil. The Ethics Committee on the Use of Animals (CEUA, Lavras, Brazil) at UFLA approved the experimental procedures under Protocol no. 067014.

Two hundred and seventy barrows were used (a cross between DanBred—DB90 females and PIC—AGPIC337 males), weaned at the age of 21 days with an initial body weight (BW) of 5.76 ± 0.52 kg. They were housed in suspended pens, with slatted floors, semiautomatic feeders, and nipple drinkers. The windows and heat lamps were used to keep the temperature of the barn close to the ideal temperature for piglets at the post-weaning age. In the first nursery week, the temperature was maintained around 26.6 ± 1.7 °C, decreasing weekly to 23.8 ± 1.1 °C in the last week of the trial. The relative humidity was 63% in the first week and 69% in the last week.

The feed was formulated to meet the nutritional requirements according to the NRC (2012), in three nursery phase periods: pre-initial 1 (21 to 31 days old), pre-initial 2 (32 to 48 days old), and initial (49 to 63 days old) (Table 1). The diets had the same composition, differing only in the participation or not of additives and an antibiotic. Levels of supplemented additives (benzoic acid—BA and essential oil—EO) were analytically verified in the trial diets as described below.

The extractable content of the additives was obtained by hot methanol extraction in a Soxhlet extraction system. The insoluble products remained in the cartridge, which was then dried and weighed. Thymol was extracted with acetone and analyzed with a gas chromatography system with a flame ionization detector. The assay was carried out with pure thymol as an external standard and p-Fluorovalerophenone as the internal standard. Benzoic acid was extracted with 50 mM sodium hydroxide as a solvent and quantified by reversed-phase HPLC–UV (272 nm) with benzoic acid as an external standard.

### 2.2. Experimental Design

The experimental design was in randomized blocks and the piglets were distributed in six treatments, with nine repetitions of five piglets each. The treatments consisted of: positive control with 40 mg/kg colistin (PC); negative control without the use of the growth promoter (NC); negative control +5 g/kg benzoic acid (BA); negative control +2 g/kg of the combination of benzoic acid and essential oils, with 1.800 g BA and 0.072 g EO (BA+EO2); negative control +3 g/kg of the combination of benzoic acid and essential oils, with 2.700 g BA and 0.108 g EO (BA+EO3); negative control +4 g/kg of the combination of benzoic acid and essential oils, with 3.600 g BA and 0.144 g EO (BA+EO4). All the additives were added in substitution of the inert material of the diets (caolin) during the feed manufacturing.

As a source of benzoic acid, VevoVitall^®^ was used, with a 99% acid concentration. The commercial product VevoWin^®^ was used for the combination of additives. It is composed of 90% benzoic acid and 3.6% of a blend of encapsulated essential oils. The blend of essential oils includes thymol, 2-methoxyphenol, and eugenol with an estimated total of 10%, and piperine and curcumin with an estimated total of 3%. DSM Produtos Nutricionais Brasil SA provided the products.

The evaluated compositions of benzoic acid and thymol for the diets were as follows: PC and NC did not present benzoic acid nor thymol, BA presented 4.96 g of benzoic acid per kg of diet and did not present thymol, BA+EO2 presented 1.61 g of benzoic acid and 2.806 mg of thymol per kg of diet, BA+EO3 presented 2.42 g of benzoic acid and 4.221 mg of thymol per kg of diet, and BA+EO4 presented 3.2 g of benzoic acid and 5.606 mg of thymol per kg of diet.

### 2.3. Experimental Procedures

The experimental period lasted 42 days. On the first experimental day, the piglets received an intramuscular dose (0.15 mL) of tulathromycin antibiotic (Draxxin^®^, *Zoetis*, Parsippany-Troy Hills, NJ, USA), for the control of respiratory diseases. According to Silveira [28], this antibiotic acts mainly in the lungs, which could minimize any other factor that is not associated with enteric challenges.

All the piglets were challenged with two doses of *Escherichia coli* F4 inoculum (LT^+^, STa^+^, and STb^+^), at a concentration of 10^6^ CFU/mL. This dose was established in a previous trial by our team [28]. They received the 1 mL dose orally on the seventh and eighth days of the experiment. The Swine Health Laboratory, at the School of Veterinary Medicine and Animal Science of University of São Paulo, isolated the *Escherichia coli* F4 (LSS-103/2008) and the Microbiology Laboratory in the UFLA Animal Science Department prepared the inoculum. The strain was grown in a culture medium for 16 h at 37 °C, and then washed in PBS, at a concentration of 10^6^ CFU/mL.

Nine piglets, one from each experimental unit (with the nearest live weight to the mean of the pen), were euthanized fourteen days after inoculation, at 42 days old. The insensibilization method used for slaughter was electronarcosis and then the exsanguination was carried out. This procedure allowed for the determination of relative organ weight, cecal content for microbiological and volatile fatty acid analyses, pH values of the GIT, intestinal histomorphometry, cell proliferation, and cholecystokinin counts.

During the second experimental week, five pigs died, one from the NC treatment, two from the BA treatment, and two from the BA+OE4 treatment, all from different experimental units. In order to correct the feed consumption and feed conversion, the procedure suggested by Sakomura and Rostagno [29] was adopted, which consists of weighing all piglets in the experimental unit as well as the feed leftovers of each experimental unit on the day of death.

### 2.4. Performance and Incidence of Diarrhea

For the performance evaluation, the piglets were weighed at 21, 35, 42, 49 and 63 days old. Average daily gain (ADG), average daily feed intake (ADFI), and feed-to-gain ratio (F:G) were calculated.

Twice a day, a trained person performed the evaluation of fecal scores by stool classification in the pen. The absence of diarrhea was determined by observing normal feces and the presence of diarrhea was determined by observing liquid and pasty feces, similar to the methodology proposed by Xiao et al. [30]. The occurrence of diarrhea was calculated by the percentage of diarrhea-positive pens compared to the total observations in one period, finding the percentage for 21 to 35, 21 to 42, 21 to 49, and 21 to 63 days old.

### 2.5. Microbiology

For the microbiological analysis, fecal samples were collected on the fourth day after inoculation and cecal content was sampled at slaughter, to compare the microbiological profile right when the challenge was established and after a period of possible recovery of the piglets. Population analyses were conducted via culture using the specific selective medium method. Ten-fold serial dilutions were made and 100 µL of each 10^−2^ to 10^−5^ dilution was mixed with 900 µL 0.9% saline seeded in plates with Rogosa medium (Sigma-Aldrich, St. Louis, MO, USA) for *Lactobacillus* spp., in BSM medium (Sigma-Aldrich, St. Louis, MO, USA) for *Bifidobacterium*, and in VRB medium (3M^TM^ Petrifilm^TM^, Sumaré, SP, Brazil) for *Escherichia coli* and total coliforms.

### 2.6. pH of the Gastrointestinal Tract and Organ Weight

After the slaughter, the pH of the content of each gastrointestinal tract segment was measured (stomach, duodenum, jejunum, ileum, and cecum) by cutting and inserting a pH meter probe (Testo 205 model—Testo do Brasil).

The pancreas, spleen, and liver of the euthanized piglets were also collected for evaluation of the relative weight of these organs. Equation (1) was used to determinate this.
(1)Organ relative weight=(organ weight÷body weight)×100

### 2.7. Volatile Fatty Acids

Following the methodology described by Playne [31], the analysis of volatile fatty acids (acetic, propionic, and butyric) was conducted based on the cecal content collected after slaughter. Briefly, 1 mL of 25% metaphosphoric acid was mixed with 5 mL of cecal content sample in a 15 mL centrifuge tube and the mixture was frozen overnight. The samples were then thawed, neutralized with 0.4 mL of 25% NaOH, and vortexed. Afterwards, 0.64 mL of 0.3 M oxalic acid was added to the samples and they were vortexed again. The samples were then centrifuged for 20 min at 3000× *g* at 4 °C and 2 mL supernatant was transferred into a gas chromatography vial for volatile fatty acids analysis. The injector and detector temperatures were, respectively, 250 and 300 °C. The furnace heating schedule started at 60 °C for 1 min, then increased to 200 °C at a heating rate of 5 °C min^−1^ and remained 1 min at the final temperature. Helium gas was used as a carrier gas and make-up at a flow rate of 1 and 29 mL min^−1^, respectively. The flame ionization detector was powered by air (300 mL min^−1^) and hydrogen (30 mL min^−1^). The chromatographic system was calibrated with standard solutions of the studied acids in the concentration range of 5–500 µg mL^−1^. The injection volume for standard solutions and samples was 1 µL. The standards used were of high purity (>99%). The acetic acid standard was obtained from Sigma-Aldrich (St. Louis, MO, USA), the propionic acid standard (>99%) was obtained from Merck (Honenbrunn, Germany), and the butyric acid standard (>99%) was obtained from Aldrich (Milwaukee, WI, USA). From these standards, working solutions containing the three acids were prepared, in different concentrations, by dilution in deionized water.

### 2.8. Intestinal Morphology

The duodenum, jejunum, ileum, and colon samples were collected, respectively, 10 cm after the pylorus, 2.0 m from the pylorus, 10 cm anterior to the cecal ileum junction, and 10 cm posterior to the cecum. They were previously washed with saline solution, fixed in Bouin solution for 12 h, dehydrated, inserted into paraffin, cut into a microtome (4 μm), and the slides were stained with hematoxylin and eosin, following the methodology proposed by Pluske et al. [32].

The duodenum, jejunum, and ileum slides were sampled with ten villi for the evaluation of villus height and with ten crypts for crypt depth, and for the colon ten crypts were sampled. They were analyzed using the OLYMPUS CX31 optical microscope with an OLYMPUS SC30 camera attached, and the Axio Vision Release 4.9 (ZEISS) analysis software was used.

### 2.9. Cellular Proliferation

New histological slides of the jejunum were prepared to evaluate cell proliferation through detection of proliferating cell nuclear antigen (PCNA), by immunohistochemistry [33]. After deparaffinization and rehydration of the slides, endogenous peroxidase activity was blocked using Peroxidase Block reagent (DakoCytomation, Fort Collins, CO, USA), and then the non-specific antibody bindings were blocked using Block Serum reagent (DakoCytomation, Fort Collins, CO, USA).

The histological sections were incubated with the anti-PCNA monoclonal antibody (PC 10—Dako^®^ A/S. Denmark) at a dilution of 1:1000 for 1 h at room temperature. Subsequently, they were incubated with the secondary polyclonal antibody (HRP Dako Envision^®^—code K401111) at a 1:1 dilution for 30 min at room temperature. For revelation, 50 μL of diaminobenzidine (DAB) was used (Dako Envision^®^, code K346811), and after immunostaining, the sections were counterstained with Carazzi hematoxylin.

Proliferation was expressed as the percentage of PCNA positive cells to total crypt cells. Five fields per sample were evaluated, nine samples per treatment, in representative areas using 400× magnification. The slides were analyzed using the OLYMPUS CX31 optical microscope with an OLYMPUS SC30 camera attached, and the ImageJ^®^ 1.41 image analysis software was used.

### 2.10. Cholecystokinin Count

Histological duodenum slides were also prepared for the cholecystokinin count (CCK) count, by immunohistochemistry [34]. After deparaffinization and rehydration of the slides, endogenous peroxidase activity and non-specific antibody bindings were blocked as described previously.

The histological sections were incubated with primary polyclonal antibody CCK-8 (T 4254—Península Lab^®^) at 1:1000 dilution for 3 h at room temperature. Subsequently, they were incubated with the secondary polyclonal antibody (HRP Dako Envision^®^—code K401111) at a 1:1 dilution for 30 min at room temperature. For revelation, 50 μL of diaminobenzidine (DAB) was used (Dako Envision^®^, code K346811), and after immunostaining, the sections were counterstained with Carazzi hematoxylin. Positive structures for CCK in the crypts were quantified. The evaluation was carried out in the same way as described for PCNA, differing only in the magnification used here, which was 200×.

### 2.11. Economic Analysis

To evaluate the viability of using each additive, an economic analysis was performed at the end of the experiment, considering different production items.

The variables analyzed were net revenue (NR), considering gross revenue from sale of piglet upon leaving the nursery (GRSP), gross cost of feed (GCF) and other production costs (OPCs), according to data from the Poultry and Swine Intelligence Center (CIAS) for Santa Catarina—Brazil, in 2014.

Other production costs were represented by cost of feed (28.83%), transport expenses (1.45%), finance charges (0.18%), piglet purchase costs (60.56%), sundry expenses (1.82%), and remuneration on breeders and pigs in stock (7.17%). Additionally, the return on investment (ROI) was analyzed, considering net gain obtained over the invested amount (gross cost of feed and other production costs). The cost per kilo of produced piglet (piglet cost/kg) was analyzed considering the costs in the nursery phase according to the pigs’ BW at 63 days old.

### 2.12. Statistical Analysis

For the performance evaluation, incidence of diarrhea, and economic analysis, the experimental units were represented by five piglets until 42 days of age and represented by four piglets after 42 days of age. For the other evaluated parameters, one piglet per pen represented the experimental units.

The results were analyzed using the Statistical Analysis System computer program (SAS Inst. Inc., Cary, NC, USA). The Shapiro–Wilk test was used to analyze the normality of the data and when they did not present this distribution, transformation was performed using PROC RANK (SAS Inst. Inc.). All variables were subjected to analysis of variance. When there was a statistical difference according to the F Test (*p* < 0.05), the Tukey test was used to compare the means.

## 3. Results

### 3.1. Performance

In the period from 21 to 35 days old, the piglets that received a combination of benzoic acid and essential oils at 3 g/kg (BA+EO3) presented a higher ADG and better F:G when compared to the piglets that received no additive (NC) (*p* < 0.05) (Table 2). In the period from 21 to 42 days old, the piglets in the BA+EO3 groups and the group that received the antibiotic (PC) presented a higher ADG when compared to the negative control piglets.

In the period from 21 to 49 days old, the BA+EO3 piglets had a greater ADFI (*p* < 0.05) compared to the NC piglets, and a greater ADG (*p* < 0.05) compared to the NC and the BA+EO2 groups. The piglets of the BA+EO3 group had a similar ADFI (*p* > 0.05) and ADG (*p* > 0.05) to the BA, BA+EO4, as well as the PC piglets.

Considering the whole experimental period, the piglets that received only 5 g/kg of benzoic acid (BA) presented a higher ADFI (*p* < 0.05) and ADG (*p* < 0.05) compared to NC and BA+EO2. A higher final BW (*p* < 0.05) was observed in this period for the piglets in the PC, BA, BA+EO3, and BA+EO4 groups compared to those in the NC group.

### 3.2. Diarrhea Incidence

No differences were observed (*p* > 0.05) for the incidence of diarrhea between treatments in the periods evaluated (Table 3). 

### 3.3. Microbiological Count

The populations of *Bifidobacterium*, *Lactobacillus* spp., *Escherichia coli*, and total coliforms were also not affected by the experimental treatments (*p* > 0.05) in the pre-killing and post-killing periods (Table 4).

### 3.4. pH, Volatile Fatty Acids, and Relative Weight of Organs

The results for pH (Table 5), production of volatile fatty acids (Table 6), and relative organ weight (Table 7) were not affected by the experimental treatments (*p* > 0.05). The experimental treatments also had no effect (*p* > 0.05) on the CCK count in the duodenum. The mean CCK number counted was 11.92 ± 1.26 units.

### 3.5. Intestinal Histology

For villus height, crypt depth, and the ratio between them, no treatment effect was observed (*p* > 0.05) in the duodenum and ileum segments (Table 8). Similarly, there was no significant difference (*p* > 0.05) for crypt depth in the colon. However, a reduction was noted (*p* < 0.05) for the villus height in the jejunum of the piglets in the BA+EO3 group compared to those in the NC and BA+EO4 groups. Stained and unstained PCNA cells in jejunum crypts were counted, but no treatment had a significant effect (*p* > 0.05). The mean cellular proliferation percentage was 46.46 ± 6.29%.

### 3.6. Economic Evaluation

The economic viability of using the treatments, considering production costs for a feeder pig operation, is shown in Table 9. 

There was a significant increase (*p* < 0.05) in gross revenue from piglet sales in the benzoic acid and the 3 g/kg benzoic acid and essential oils combination groups compared to the negative control group piglets. There was also a significant increase (*p* < 0.05) in net revenue when the 3 g/kg and 4 g/kg benzoic acid and additive combination treatments were used compared to the negative control. Return on investment was better (*p* < 0.05) for the benzoic acid and 3 g/kg and 4 g/kg combination groups compared to the negative control group without differing from the positive control. The cost per kilo of piglet produced was lower (*p* < 0.05) for the 3 g/kg and 4 g/kg benzoic acid combination groups compared to the negative control group and equal (*p* > 0.05) in comparison with the positive control. The 3 g/kg dose has the best cost/benefit ratio.

## 4. Discussion

The dose of inoculum used in this trial was based on a previous trial by our group [28], however, it was low if compared to other trials [35,36,37]. Nevertheless, the challenge seemed to be effective at increasing the diarrhea among the piglets as higher values were observed for the first period evaluated. However, to better elucidate its effect, one treatment without the challenge should be provided, and this is thus a limitation of our study. In future trials, a treatment with it should be included to ensure the efficacy of the challenge and the beneficial effects of the additives. 

The use of benzoic acid (BA) generally results in benefits in body weight gain, ADFI, and feed conversion of piglets after weaning [19,28,38,39,40]. These benefits may be related with the fact that BA administration promotes the production and activation of digestive enzymes such as trypsin, lipase, amylase, maltase, sucrose, and lactase. In addition, BA improves the absorption of nutrients [41]. The use of essential oils also favors increased ADFI, greater piglet weight [42,43], and better feed conversion [44,45]. Studies show similarly significant results among piglets supplemented with an antibiotic or essential oils in terms of weight gain and feed conversion, highlighting the possibility of substituting antibiotics [46,47].

The better performance at 35 days old resulting from the combination of additives at 3 g/kg (BA+EO3) compared to the negative control was not accompanied by greater ADFI. The animals in the BA+EO3 group probably presented better gastrointestinal tract conditions and a better rate of nutrient digestibility. As noted by Zhang et al. [48], providing 3 g/kg of benzoic acid +0.1 g/kg of essential oils was efficient in increasing apparent total digestibility of dry matter and nitrogen in piglets at twenty-one days of age in the nursery. Essential oils and benzoic acid are regarded as stimulants of digestive enzyme secretion, which may improve nutrient absorption and feed conversion [26,49,50]. Moreover, this combination improves the intestinal mucosal barrier integrity in enterotoxigenic challenged pigs [51].

In the present study, the use of the thymol, 2-methoxyphenol, eugenol, piperine, and curcumin components in combination with benzoic acid may have resulted in the greater weight gain for the BA+EO3 piglets compared to the negative control group, due to the joint action of the essential oils. In contrast, the inclusion of only one essential oil associated with benzoic acid may not improve weight gain and ADFI in this period compared with the negative control group, as observed by Diao et al. [52]. It is known that the action of the major component of a plant can be potentiated by secondary components, causing a synergistic effect, in the form of blends, unlike when a component is used separately [46,53,54].

Although the combinations of additives at 3 g/kg (BA+EO3) and 4 g/kg (BA+EO4) positively affected the performance variables in some periods evaluated, the combination at the lower level of 2 g/kg (BA+EO2) showed inferior performance. This result may be related to the lower amount of benzoic acid (1.8 g/kg) and essential oils (0.072 g/kg) lacking a sufficient concentration and synergy to improve the performance variables, thus contrasting with some of the positive results for these variables when these additives are used. In the literature, superior performance characteristics for piglets receiving benzoic acid resulted from a linear effect of the addition at the following levels: 0, 2.5, 5.0, and 7.5 g/kg [55]. In addition, positive effects of essential oils on ADFI and weight gain in piglets can be observed with the inclusions of 0.1 and 0.15 g/kg [22], which are greater levels than the BA+EO2 group received.

Considering the performance of piglets throughout the nursery phase, benzoic acid and combinations with EO at the levels of 3 g/kg and 4 g/kg favored a greater final BW in the piglets, similarly to what was observed for the piglets receiving the antibiotic. Their performance was improved, possibly due to better nutrient digestibility, as well as increased bacterial diversity and richness caused by organic acid and the essential oils [53,56,57,58].

The treatments did not influence the incidence of diarrhea among the piglets, as also observed by Halas et al. [40] and Pu et al. [51] in pigs challenged with enterotoxigenic *E. coli* and supplemented with benzoic acid. The distribution of diarrhea among the piglets, regardless of the treatment, is concentrated in the first week after weaning, as is typical of this period due to the stress and physiological unpreparedness for the food change. It remained high during the second week of the experiment, because on the seventh and eighth day of nursery, the piglets were inoculated with the highly pathogenic bacterial strain of *Escherichia coli F4*.

Just as the treatments presented no effect on diarrhea, the microbiological profile was also not altered by them. The results differ from studies such as that of Diao et al. [17], in which the use of benzoic acid lowered the *Escherichia coli* count and raised the count of *Bifidobacterium* spp. in the ileum of piglets. The plant extracts are able to inhibit this mechanism and can modulate the biofilm formation and the motility of *E. coli* [59,60]. We speculate that in this study, the benzoic acid and essential oils may have been inhibitors of quorum sensing, reducing the communication of bacterial populations and improving the performance of the piglets, without altering the count of these microorganisms.

As volatile fatty acids are the major end products of bacterial metabolism in the large intestine of swine, the effect of organic acids or essential oils on augmenting bacterial counts can increase this production. However, in this study the bacterial count remained the same in all treatments, which may have accounted for the similar VFA concentrations between treatments.

The use of benzoic acid or the combination with EO also did not reduce the pH of the evaluated gastrointestinal tract segments, supporting the suggestion that the effect of benzoic acid is not necessarily linked to a lower pH in the segments [61]. Its high dissociation constant (pKa = 4.21) hinders its dissociation and acidification of the medium. A similar result was observed by Halas et al. [40], when piglets supplemented with benzoic acid presented no change in gastric pH.

The experimental treatments also did not alter the cholecystokinin count (CCK) in the duodenum. CCK is released post-prandially in response to saturated fat, long-chain fatty acids, amino acids, and peptides that result from protein digestion. It is able to promote constriction of the pyloric sphincter, slowing gastric emptying, and is the main stimulus for the release of pancreatic enzymes and bile in the small intestine [62]. This hormone can be secreted by the I cells of the duodenum and this occurs more easily in acidic conditions [34,63]. Considering this hormone’s susceptibility to pH, the absence of a change in duodenum pH caused by the treatments may have influenced the absence of a significant difference in the CCK count.

Morphological alterations of the intestine may be a consequence of a natural process in pigs older than three weeks of age, where the epithelium undergoes periodic renewal, with migration of crypt cells to the end of the villi. Morphological and functional differentiations occur in this migration [64] and an accelerated renewal rate may result in villi with immature cells and reduced enzymatic activity [65,66,67]. Thus, the higher villus height in the jejunum of piglets in the negative control group compared to the group that received the combination of additives at a medium level may be indicative of the presence of “immature” cells coming from the crypt, given that these piglets showed much inferior performance in the period from 21 to 49 days of age, which includes the segment sample collection date. 

On the other hand, despite a lower villus height in the jejunum of the BA+EO3 group, nutrients were probably absorbed more efficiently, with less damage to the villi, culminating in better performance. However, none of the treatments resulted in an increase or decrease in cellular proliferation in the jejunal crypts of the piglets, as evaluated by immunohistochemistry.

As changes in the intestine of piglets are constant after weaning, some organs may also be susceptible to changes in their weights. However, the experimental treatments did not influence organ size. This result is similar to that of an evaluation of essential oils in chickens, in which there was no increase in the weight of the pancreas, liver, and small intestine, even though there was a change in ADFI in some periods [68]. According to Rao and McCracken [69], the weight of organs may vary according to the energy content and/or protein content of the diet. This fact may clarify the similarity between the relative weights of the organs presented in this research, because the diets used were isoproteic and isocaloric.

Given the positive results for the performance variables evaluated in this study, it is important to evaluate the economic return that can be obtained by investing in the combined use of benzoic acid and essential oils in the nursery phase. The cost varied for feed in each treatment; however, when considering how much feed the pig consumed during the experiment and its respective value, there was no increase in expense between including an antibiotic, benzoic acid, or benzoic acid and essential oil combinations in the baseline diet.

Considering the production costs, which are fixed, for any of the treatments evaluated, the use of benzoic acid or the combinations of this acid and essential oils at 3 g/kg and 4 g/kg resulted in greater net revenue than with the use of a baseline diet without additives. Hence, the return achieved from the amount invested in the piglets for experimental treatments BA, BA+EO3, and BA+EO4 was higher than when the baseline feed without additive was used, aside from being equal in terms of the investment made in piglets that received feed with colistin. The cost per kilo of piglet produced also shows that the BA, BA+EO3, and BA+EO4 treatments are more advantageous, given that the values were lower than the cost per kilo of piglet in the treatment with baseline feed and equal to those in the treatment with colistin.

## 5. Conclusions

The use of benzoic acid alone (5 g/kg) or combined with essential oils (3 g/kg and 4 g/kg), as evaluated in this study, improves the performance of piglets in the nursery phase and can be used with the same result as the use of the antibiotic colistin.

## Figures and Tables

**Table 1 animals-10-01978-t001:** Centesimal composition and calculated nutritional values of the diets used in the different phases of the experiment.

Ingredients (g/kg, as-Fed Basis)	Pre-Initial 1	Pre-Initial 2	Initial
Corn	245.00	395.00	615.00
Soybean meal 45%	150.00	200.00	270.00
Micronized whole soybeans	90.00	17.50	70.00
Soy protein concentrate	60.00	62.50	-
Starch	234.49	158.21	07.57
Whey powder	120.97	62.50	-
Dairy product, 38% lactose ^1^	60.00	62.50	-
DL-methionine	2.27	1.87	1.01
L-tryptophan	0.52	0.40	0.08
L-threonine	1.85	1.58	0.91
L-lysine	5.00	4.34	3.12
L-valine	1.14	0.71	-
Phytase	0.10	0.10	0.10
Premix Vit/Min ^2^	10.00	10.00	10.00
Salt	2.00	2.50	4.00
Limestone	-	-	5.79
Dicalcium phosphate	5.42	7.04	6.68
Calcium sulfate	5.50	7.50	-
Antioxidant ^3^	0.25	0.25	0.25
Flavoring ^4^	0.50	0.50	0.50
Caolin ^5^	5.00	5.00	5.00
Calculated values *n* (g/kg, unless specified)			
Metabolizable energy, MJ/kg	15.5	15.1	13.6
Crude protein	214.8	190.8	204.4
Total lysine	15.1	14.4	13.2
Digestible lysine	14.5	13.9	11.7
Ca	7.3	6.3	6.0
P	5.2	4.7	4.5
Lactose	107.5	67.5	-

^1^ Commercial product Nuklospray E50. ^2^ Composition/kg of product: 12 mg Cu as CuSO_4_; 80 mg Fe as FeSO_4_); 1 mg I as Ca(IO_3_)_2_; 40 mg Mn as MnO_2_; 36 mg Se as NaSeO_3_; 110 mg Zn as ZnO; vit. A (6875.00 I.U.); vit. D3 (1505.00 I.U.); vit. E (40.00 mg); vit. K3 (3.07 mg); vit. B1 (1.00 mg); vit. B2 (3.13 mg); vit. B6 (2.00 mg); vit. B12 (0.02 mg); niacin (30.00 mg); folic acid (0.30 mg); pantothenic acid (15.00 mg); biotin (0.10 mg); choline (200.97 mg). ^3^ Commercial product Banox E. ^4^ Commercial product Luctarom Lactantes. ^5^ Inert matter composed of hydrated aluminum silicates.

**Table 2 animals-10-01978-t002:** Effect of experimental diets on the performance of piglets challenged with *E. coli* F4.

Variables	PC	NC	BA	BA+EO2	BA+EO3	BA+EO4	SEM	*p-*Value
21 to 35 days of age								
BW	5.77	5.76	5.77	5.76	5.76	5.76	0.001	0.135
ADFI	0.22	0.20	0.22	0.21	0.23	0.21	0.010	0.232
ADG	0.149 ^a,b^	0.112 ^c^	0.143 ^a,b,c^	0.126 ^b,c^	0.164 ^a^	0.132 ^a,b,c^	0.008	0.001
F:G	1.47 ^a,b^	1.86 ^a^	1.57 ^a,b^	1.73 ^a,b^	1.42 ^b^	1.63 ^a,b^	0.089	0.014
21 to 42 days of age								
ADFI	0.34	0.29	0.34	0.30	0.34	0.32	0.013	0.231
ADG	0.22 ^a^	0.17 ^b^	0.20 ^a,b^	0.19 ^a,b^	0.23 ^a^	0.19 ^a,b^	0.010	0.003
F:G	1.57	1.79	1.56	1.59	1.50	1.73	0.077	0.092
21 to 49 days of age								
ADFI	0.409 ^a,b^	0.355 ^b^	0.415 ^a^	0.365 ^a,b^	0.414 ^a^	0.407 ^a,b^	0.014	0.005
ADG	0.27 ^a,b^	0.21 ^c^	0.26 ^a,b^	0.22 ^b,c^	0.28 ^a^	0.25 ^a,b,c^	0.012	0.001
F:G	1.55	1.72	1.62	1.73	1.48	1.64	0.075	0.129
21 to 63 days of age								
ADFI	0.63 ^a,b^	0.57 ^b^	0.67 ^a^	0.59 ^b^	0.63 ^a,b^	0.63 ^a,b^	0.016	0.002
ADG	0.408 ^a,b^	0.357 ^c^	0.421 ^a^	0.377 ^b,c^	0.419 ^a,b^	0.407 ^a,b^	0.010	<0.001
F:G	1.54	1.61	1.58	1.56	1.51	1.56	0.028	0.200
BW	22.96 ^a^	20.78 ^b^	23.45 ^a^	21.70 ^a,b^	23.38 ^a^	22.98 ^a^	0.416	<0.001

PC, positive control with 40 mg/kg colistin; NC, negative control without the use of the growth promoter; BA, negative control +5 g/kg benzoic acid; BA+EO2, negative control +2 g/kg of the combination of benzoic acid and essential oils; BA+EO3, negative control +3 g/kg of the combination of benzoic acid and essential oils; BA+EO4, negative control +4 g/kg of the combination of benzoic acid and essential oils. BW, body weight (kg); ADFI, average daily feed intake (kg/day); ADG, average daily gain (kg/day); F:G, feed-to-gain ratio (kg/kg); SEM, standard error of the mean. Means followed by different letters are significantly different according to the Tukey test (*p* < 0.05).

**Table 3 animals-10-01978-t003:** Effect of experimental diets on the incidence of diarrhea in piglets challenged with *E. coli* F4.

Variables	PC	NC	BA	BA+EO2	BA+EO3	BA+EO4	SEM	*p-*Value	
21 to 35 days of age								
ID, %	24.1	37.5	34.6	27.4	29.3	29.0	4.893	0.432
21 to 42 days of age								
ID, %	21.9	31.1	28.6	27.9	26.8	22.3	3.313	0.267
21 to 49 days of age								
ID, %	19.1	28.4	25.6	24.0	27.1	21.9	2.933	0.207
21 to 63 days of age								
ID, %	13.3	19.4	17.5	17.3	15.8	16.0	2.175	0.316

PC, positive control with 40 mg/kg colistin; NC, negative control without the use of the growth promoter; BA, negative control +5 g/kg benzoic acid; BA+EO2, negative control +2 g/kg of the combination of benzoic acid and essential oils; BA+EO3, negative control +3 g/kg of the combination of benzoic acid and essential oils; BA+EO4, negative control +4 g/kg of the combination of benzoic acid and essential oils. ID, incidence of diarrhea (%); SEM, standard error of the mean.

**Table 4 animals-10-01978-t004:** Effect of experimental diets on the microbial concentration (CFU/g) in feces collected at 33 days of age (pre-killing) and on the cecal content collected at 42 days of age (post-killing) from piglets challenged with *E. coli* F4.

Variables	PC	NC	BA	BA+EO2	BA+EO3	BA+EO4	SEM	*p-*Value
33 days of age (pre-killing)								
*Bifidobacterium*	3.3 × 10^8^	1.4 × 10^8^	2.4 × 10^8^	1.4 × 10^8^	3.2 × 10^8^	1.7 × 10^8^	1.140	0.273
*Lactobacilli* spp.	3.7 × 10^8^	2.1 × 10^8^	2.5 × 10^8^	2.0 × 10^8^	2.7 × 10^8^	1.4 × 10^8^	0.579	0.253
*Escherichia coli*	1.9 × 10^7^	1.3 × 10^8^	7.5 × 10^5^	2.9 × 10^6^	4.6 × 10^6^	5.7 × 10^7^	2.819	0.143
Total Coliforms	1.9 × 10^7^	1.6 × 10^8^	1.3 × 10^6^	3.1 × 10^6^	4.3 × 10^6^	5.7 × 10^7^	1.854	0.245
42 days of age (post-killing)								
*Bifidobacterium*	2.7 × 10^7^	1.1 × 10^7^	2.1 × 10^7^	2.3 × 10^7^	1.4 × 10^7^	2.7 × 10^7^	0.707	0.971
*Lactobacilli* spp.	1.3 × 10^7^	8.8 × 10^6^	1.3 × 10^7^	6.8 × 10^6^	1.8 × 10^7^	1.1 × 10^7^	1.043	0.811
*Escherichia coli*	2.0 × 10^5^	2.1 × 10^6^	5.2 × 10^5^	1.5 × 10^9^	1.2 × 10^6^	1.2 × 10^6^	3.128	0.523
Total Coliforms	3.1 × 10^5^	2.5 × 10^6^	8.1 × 10^5^	1.5 × 10^9^	1.6 × 10^6^	1.2 × 10^6^	2.320	0.457

PC, positive control with 40 mg/kg colistin; NC, negative control without the use of the growth promoter; BA, negative control +5 g/kg benzoic acid; BA+EO2, negative control +2 g/kg of the combination of benzoic acid and essential oils; BA+EO3, negative control +3 g/kg of the combination of benzoic acid and essential oils; BA+EO4, negative control +4 g/kg of the combination of benzoic acid and essential oils. SEM, standard error of the mean.

**Table 5 animals-10-01978-t005:** Effect of experimental diets on the pH of piglets challenged with *E. coli* F4.

Variables	PC	NC	BA	BA+EO2	BA+EO3	BA+EO4	SEM	*p-*Value
Stomach	3.7	3.1	3.5	3.4	3.2	3.7	0.780	0.518
Duodenum	5.7	5.7	5.7	5.3	5.3	5.5	0.745	0.724
Jejunum	5.9	6.2	6.0	5.9	6.1	5.9	0.429	0.757
Ileum	6.3	6.2	6.3	6.2	6.2	6.3	0.511	0.949
Cecum	5.7	5.7	5.7	5.6	5.6	5.6	0.231	0.642

PC, positive control with 40 mg/kg colistin; NC, negative control without the use of the growth promoter; BA, negative control +5 g/kg benzoic acid; BA+EO2, negative control +2 g/kg of the combination of benzoic acid and essential oils; BA+EO3, negative control +3 g/kg of the combination of benzoic acid and essential oils; BA+EO4, negative control +4 g/kg of the combination of benzoic acid and essential oils. SEM, standard error of the mean.

**Table 6 animals-10-01978-t006:** Effect of experimental diets on the production of volatile fatty acids (mMol/g) in the cecum of piglets challenged with *E. coli* F4.

Variables	PC	NC	BA	BA+EO2	BA+EO3	BA+EO4	SEM	*p-*Value
Acetic	606.3	594.5	577.0	656.4	512.0	618.5	157.20	0.783
Propionic	337.8	363.8	297.7	329.2	300.4	298.7	93.27	0.649
Butyric	126.2	157.2	155.6	169.7	155.7	144.1	55.53	0.595

PC, positive control with 40 mg/kg colistin; NC, negative control without the use of the growth promoter; BA, negative control +5 g/kg benzoic acid; BA+EO2, negative control +2 g/kg of the combination of benzoic acid and essential oils; BA+EO3, negative control +3 g/kg of the combination of benzoic acid and essential oils; BA+EO4, negative control +4 g/kg of the combination of benzoic acid and essential oils. SEM, standard error of the mean.

**Table 7 animals-10-01978-t007:** Effect of experimental diets on the relative weight of organs (%) in piglets challenged with *E. coli* F4.

Variables	PC	NC	BA	BA+EO2	BA+EO3	BA+EO4	SEM	*p-*Value
Pancreas	0.24	0.24	0.22	0.24	0.23	0.21	0.031	0.419
Spleen	0.21	0.29	0.23	0.21	0.23	0.21	0.110	0.967
Liver	2.96	3.16	3.13	3.07	3.15	3.12	0.347	0.861

PC, positive control with 40 mg/kg colistin; NC, negative control without the use of the growth promoter; BA, negative control +5 g/kg benzoic acid; BA+EO2, negative control +2 g/kg of the combination of benzoic acid and essential oils; BA+EO3, negative control +3 g/kg of the combination of benzoic acid and essential oils; BA+EO4, negative control +4 g/kg of the combination of benzoic acid and essential oils. SEM, standard error of the mean.

**Table 8 animals-10-01978-t008:** Effect of experimental diets on the intestinal morphology of piglets challenged with *E. coli* F4.

Variables	PC	NC	BA	BA+EO2	BA+EO3	BA+EO4	SEM	*p-*Value
*Duodenum*								
VH	265.3	282.7	285.0	262.6	292.4	293.8	32.12	0.108
CD	203.2	210.8	212.6	201.0	199.3	208.4	26.99	0.842
VH:CD	1.4	1.4	1.4	1.4	1.5	1.5	0.17	0.382
*Jejunum*								
VH	233.1 ^a,b^	255.8 ^a^	242.4 ^a,b^	247.1 ^a,b^	220.1 ^b^	271.0 ^a^	35.99	0.012
CD	174.1	197.8	170.6	189.6	175.6	193.5	24.61	0.141
VH:CD	1.4	1.3	1.5	1.3	1.3	1.5	0.21	0.412
*Ileum*								
VH	229.9	217.7	217.5	231.9	210.6	203.4	44.00	0.942
CD	192.8	215.1	198.1	212.6	192.3	215.7	28.77	0.200
VH:CD	1.3	1.1	1.2	1.1	1.1	1.0	0.28	0.585
*Colon*								
CD	351.2	312.0	334.9	364.4	330.6	321.4	53.67	0.417

PC, positive control with 40 mg/kg colistin; NC, negative control without the use of the growth promoter; BA, negative control +5 g/kg benzoic acid; BA+EO2, negative control +2 g/kg of the combination of benzoic acid and essential oils; BA+EO3, negative control +3 g/kg of the combination of benzoic acid and essential oils; BA+EO4, negative control +4 g/kg of the combination of benzoic acid and essential oils. VH, villus height (µm); CD, crypt depth (µm); VH:CD, villus height/crypt depth ratio; SEM, standard error of the mean. Means followed by different letters are significantly different according to the Turkey test (*p <* 0.05).

**Table 9 animals-10-01978-t009:** Economic evaluation of using different experimental treatments.

Variables	PC	NC	BA	BA+EO2	BA+EO3	BA+EO4	SEM	*p-*Value
GRSP	45.5 ^a,b^	41.1 ^b^	46.4 ^a^	42.6 ^a,b^	46.5 ^a^	46.0 ^a,b^	3.688	0.005
GFC	7.4	6.8	7.6	6.8	7.5	7.4	0.740	0.087
OPC	18.1	18.1	18.1	18.1	18.1	18.1	0.000	1.000
NR	20.0 ^a,b^	16.3 ^b^	20.7 ^a^	17.7 ^a,b^	21.0 ^a^	20.4 ^a^	3.107	0.005
ROI	78.3 ^a,b^	65.5 ^b^	80.6 ^a^	70.8 ^a,b^	81.8 ^a^	80.1 ^a^	3.357	0.006
Piglet cost/kg	1.11 ^a,b^	1.20 ^a^	1.10 ^b^	1.16 ^a,b^	1.09 ^b^	1.10 ^b^	0.022	0.005

PC, positive control with 40 mg/kg colistin; NC, negative control without the use of the growth promoter; BA, negative control +5 g/kg benzoic acid; BA+EO2, negative control +2 g/kg of the combination of benzoic acid and essential oils; BA+EO3, negative control +3 g/kg of the combination of benzoic acid and essential oils; BA+EO4, negative control +4 g/kg of the combination of benzoic acid and essential oils. GRSP, gross revenue from the sale of piglets (USD); GFC, gross feed cost (USD); OPC, other production costs (USD); NR, net revenue (USD), ROI, return in investment (%); piglet cost/kg, cost per kilo of piglet produced (USD); SEM, standard error of the mean. Means followed by different letters are significantly different according to the Turkey test (*p <* 0.05).

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
