# Peer review of "Benzoic Acid Combined with Essential Oils Can Be an Alternative to the Use of Antibiotic Growth Promoters for Piglets Challenged with *E. coli* F4"

_animals, 2020, doi:10.3390/ani10111978_

Round 1
Reviewer 1 Report
Dear authors,
Thank you for your work in this line of research and for submitting this manuscript for publication. Alternatives to antibiotics as growth promoters in pigs has been widely researched in the past few year, especially considering the severe restriction or total ban on the use of antibiotics in swine production. It makes your manuscript timely and relevant to the current swine industry. The general structure of this manuscript is clearly laid out and all the key elements are included. In addition, the experimental design is suited to target this question and the language is for the most part appropriate and understandable. However, some parts can be improved to maximize the quality of this manuscript. Some revisions are recommended and detailed below:
- Abstract:
P shall be P (Italic). Check this out in all the text.
Common abbreviations for performance variables in swine production include: ADG, average daily gain; ADFI, average daily feed; BW, body weight; G:F, gain-to-feed ratio. Considering that these abbreviations are widely recognized and used in scientific language, it could be adopted in this manuscript. Please, consider this comment throughout the entire manuscript.
Why BA is not considered in the conclusion of the abstract? Please check and rewrite a consistent conclusion.
- Introduction:
Line 47 “…to maintain the animal’s performance [2]” If the context of this sentence is swine, why use the word animals? In this case and many other cases throughout the manuscript, it is recommended to use specifically the word pigs/piglets instead of animal/animals… review all the text accordingly.
Line 70-73 More details on the effects of benzoic acid and essential oils (alone and in combination) from the current literature can be better explored in this session (levels of inclusion, metrics for the improvement: i.e the supplementation improved ADG by 10%).
- Material and Methods:
Line 141-1422 Why this concentration/dose? … maybe this should be referenced or discussed in the discussion session.
Line 163-167 Please provide more detailed information about fecal scores/stool classification (include pictures or provide reference).
- Results:
Table 2. Please indicate that the periods refer to days of age. Please, consider this comment for the other tables.
Table 4. Please indicate what does it mean Pre-killing/Post-killing (specify the days).
- Discussion:
Need discussion about the effectiveness of the challenge with Escherichia coli F4.
Please consider the following articles to enrich the discussion:
Benzoic Acid Used as Food and Feed Additives Can Regulate Gut Functions (Biomed Res Int. 2019 Feb 26;2019:5721585. doi: 10.1155/2019/5721585).
The use of an alternative feed additive, containing benzoic acid, thymol, eugenol, and piperine, improved growth performance, nutrient and energy digestibility, and gut health in weaned piglets (J Anim Sci. 2020 May 1;98(5):skaa119. doi: 10.1093/jas/skaa119).
Protective Effects of Benzoic Acid, Bacillus Coagulans, and Oregano Oil on Intestinal Injury Caused by Enterotoxigenic Escherichia coli in Weaned Piglets (Biomed Res Int. 2018 Aug 27; 2018:1829632. doi: 10.1155/2018/1829632. eCollection 2018).
Effects of benzoic acid, Bacillus coagulans and oregano oil combined supplementation on growth performance, immune status and intestinal barrier integrity of weaned piglets (Anim Nutr. 2020 Jun; 6(2):152-159. doi: 10.1016/j.aninu.2020.02.004. Epub 2020 Mar 31).
Author Response
Dear Reviewer #1, we gratefully acknowledge your thorough evaluation and valuable comments. We do believe that it have contributed to improve the quality of our manuscript. All the changes in the manuscript made from your comments are highlighted in yellow. Other general changes made by the authors, such as English or update results are standardized with the green color. In addition, changes proposed by more than one reviewer are marked by the color of the first reviewer who suggested such changes, indicating the lines where the changes were made.
Comments to author:
Comment 1: “P shall be P (Italic). Check this out in all the text.”
We revised all the paper the changes are highlighted with the yellow color.
Comment 2: “Common abbreviations for performance variables in swine production include: ADG, average daily gain; ADFI, average daily feed; BW, body weight; G:F, gain-to-feed ratio. Considering that these abbreviations are widely recognized and used in scientific language, it could be adopted in this manuscript. Please, consider this comment throughout the entire manuscript.”
We revised all the paper the changes are highlighted with the yellow color.
Why BA is not considered in the conclusion of the abstract? Please check and rewrite a consistent conclusion.
The conclusion of the abstract was changed. Line 37.
Introduction:
Line 47 “…to maintain the animal’s performance [2]” If the context of this sentence is swine, why use the word animals? In this case and many other cases throughout the manuscript, it is recommended to use specifically the word pigs/piglets instead of animal/animals… review all the text accordingly.
We revised all the paper and changed the word animals by pigs or piglets highlighted with the yellow color.
Line 70-73 More details on the effects of benzoic acid and essential oils (alone and in combination) from the current literature can be better explored in this session (levels of inclusion, metrics for the improvement: i.e the supplementation improved ADG by 10%).
We include information lines 72-74.
Material and Methods:
Line 141-1422 Why this concentration/dose? … maybe this should be referenced or discussed in the discussion session.
We included a reference why we chose this dose it is on line 143. It was also discussed on the discussion section.
Line 163-167 Please provide more detailed information about fecal scores/stool classification (include pictures or provide reference).
It was included one reference (Xiao et al., 2014) on lines 165-166
Results:
Table 2. Please indicate that the periods refer to days of age. Please, consider this comment for the other tables.
The information that is related to the age of the animals was included in all tables.
Table 4. Please indicate what does it mean Pre-killing/Post-killing (specify the days).
It was included
Discussion:
Need discussion about the effectiveness of the challenge with Escherichia coli F4.
It was added on lines 320-324.
Please consider the following articles to enrich the discussion:
Benzoic Acid Used as Food and Feed Additives Can Regulate Gut Functions (Biomed Res Int. 2019 Feb 26;2019:5721585. doi: 10.1155/2019/5721585).
It was added.
The use of an alternative feed additive, containing benzoic acid, thymol, eugenol, and piperine, improved growth performance, nutrient and energy digestibility, and gut health in weaned piglets (J Anim Sci. 2020 May 1;98(5):skaa119. doi: 10.1093/jas/skaa119).
It was added.
Protective Effects of Benzoic Acid, Bacillus Coagulans, and Oregano Oil on Intestinal Injury Caused by Enterotoxigenic Escherichia coli in Weaned Piglets (Biomed Res Int. 2018 Aug 27; 2018:1829632. doi: 10.1155/2018/1829632. eCollection 2018).
It was added.
Effects of benzoic acid, Bacillus coagulans and oregano oil combined supplementation on growth performance, immune status and intestinal barrier integrity of weaned piglets (Anim Nutr. 2020 Jun; 6(2):152-159. doi: 10.1016/j.aninu.2020.02.004. Epub 2020 Mar 31).
It was added.

Reviewer 2 Report
General comments:
The manuscript is interesting and within the journal’s scope. It has been prepared quite well. but English quality must be improved. The Introduction is lacking information on a dose of essential oils used in other studies. This would justified the level of supplementation. In this type of studies, there should be a real control group of health, unchallenged piglets. Could Authors explain why such group was not included in the trial? This group would show whether dietary supplementation with benzoic acid and EO allows to obtain results similar to those of health piglets.
Minor comments:
Simple summary and Abstract: it should be mentioned what essential oils were used in the study.
l. 25: replace “wear of” by “impairment”
l. 38: an economically (not and)
l. 42: The GIT in general is a complex environment, so please rewrite the first sentence.
l. 52-54: …a highly proliferating pathogenic bacterial strain…
l. 62-63: In my opinion, the order of events is wrong. First – overuse of colistin, second – development of resistance in GNB
l. 126-130: Information on how additives were added to diets should be given. Were supplements given on a top of a diet, were they mixed with feed before feeding?
l. 131: evaluated
l. 131-135: methods of analyses of benzoic acid and essential oils in feeds should be described
l. 177: pH
l. 182-186: How were samples prepared for GC? What temperature program was applied? What was used as an internal standard?
l. 210: magnification of 400x
l. 244-247: A pen with five or four animals was an experimental unit.
l. 249: normality
l. 265: It cannot be said that something “was significantly equal”, please correct.
l. 318: One decimal place does not allow to see that the 3 g/kg dose has the best cost/benefit ratio.
l. 334:treated animals
l. 341: replace “inclusions have been positive on” by “positively affected”
l. 343: delete “this treatment have”
l. 346: resulted from a linear effect of the addition at levels…
l. 351: combinations with EO at…
l. 352: due to
l. 358: It remained high…
l. 362: The results differ..
l. 363: Bifidobacterium spp
l. 364-367: its unclear and seems to not be related to obtained results. Please rewrite or delete.
l. 368-371: The obtained results do not show this. It is a speculation.
l. 376: …or the combination with EO
l. 398-403: this paragraph should be rewritten because of the quality of English. What is meant by “less wear on the villi? Less damage? …and leading/resulting in better performance…
l. 402: jejunal crypts…as evaluated by…
l. 404: As changes…
l. 411: isoprotein and isocaloric
l. 423: …of produced piglet
Author Response
Dear Reviewer #2, we gratefully acknowledge your thorough evaluation and valuable comments. We do believe that it has contributed to improve the quality of our manuscript. All the changes in the manuscript made from your comments are highlighted in blue. Other general changes made by the authors, such as English or update results are standardized with the green color. In addition, changes proposed by more than one reviewer are marked by the color of the first reviewer who suggested such changes, indicating the lines where the changes were made.
The manuscript is interesting and within the journal’s scope. It has been prepared quite well. but English quality must be improved. The Introduction is lacking information on a dose of essential oils used in other studies. This would justified the level of supplementation. In this type of studies, there should be a real control group of health, unchallenged piglets. Could Authors explain why such group was not included in the trial? This group would show whether dietary supplementation with benzoic acid and EO allows to obtain results similar to those of health piglets.
Thanks for the comments, the English was reviewed. We included some doses of essential oils on the introduction too. We agree that a control group with unchallenged pigs should have been used, unfortunately, when we first draw the trial that was not considered, as we had limited number of replicates and the major objective was to evaluate the different combinations of the additives, In future trial it would be considered, we add discussion regarding this limitations of our study.
Minor comments:
Simple summary and Abstract: it should be mentioned what essential oils were used in the study.
It was included on lines 19, 25-26.
- 25: replace “wear of” by “impairment”
It was changed on line 26.
- 38: an economically (not and)
It was corrected on line 39.
- 42: The GIT in general is a complex environment, so please rewrite the first sentence.
It was changed.
- 52-54: …a highly proliferating pathogenic bacterial strain…
It was changed. Line 54
- 62-63: In my opinion, the order of events is wrong. First – overuse of colistin, second – development of resistance in GNB
It was changed. Line 63-64
- 126-130: Information on how additives were added to diets should be given. Were supplements given on a top of a diet, were they mixed with feed before feeding?
It was added. Line 126-128
- 131: evaluated
It was added. Line 134.
- 131-135: methods of analyses of benzoic acid and essential oils in feeds should be described
It is described on lines 112-118.
- 177: pH
It was changed. Line 180
- 182-186: How were samples prepared for GC? What temperature program was applied? What was used as an internal standard?
It was included. Lines 187-202
- 210: magnification of 400x
It was changed. Line 227
- 244-247: A pen with five or four animals was an experimental unit.
Until 42 days of age a pen with five piglets was the experimental unit, after that a pen with four piglets was the experimental unit.
- 249: normality
It was changed. Line 265
- 265: It cannot be said that something “was significantly equal”, please correct.
It was changed. Lines 280-281
- 318: One decimal place does not allow to see that the 3 g/kg dose has the best cost/benefit ratio.
It was corrected on table 9.
- 334: treated animals
It was changed. Lines 361
- 341: replace “inclusions have been positive on” by “positively affected”
It was changed. Lines 367-368
- 343: delete “this treatment have”
It was deleted. Lines 370.
- 346: resulted from a linear effect of the addition at levels…
It was changed. Lines 373
- 351: combinations with EO at…
It was changed. Lines 377
- 352: due to
It was changed. Lines 379
- 358: It remained high…
It was changed. Lines 385
- 362: The results differ..
It was changed. Lines 389
- 363: Bifidobacterium spp
It was changed. Lines 390
- 364-367: its unclear and seems to not be related to obtained results. Please rewrite or delete.
It was deleted.
- 368-371: The obtained results do not show this. It is a speculation.
The phrase was rewritten. Line 392
- 376: …or the combination with EO
It was changed. Lines 399
- 398-403: this paragraph should be rewritten because of the quality of English. What is meant by “less wear on the villi? Less damage? …and leading/resulting in better performance…
It was rewritten. Lines 421-424
- 402: jejunal crypts…as evaluated by…
It was changed. Lines 424
- 404: As changes…
It was changed. Lines 425
- 411: isoprotein and isocaloric
It was changed. Lines 432
- 423: …of produced piglet
It was changed. Lines 444

Reviewer 3 Report
The research question is important. It is necessary to investigate alternative additives to antibiotics (mainly for Colistin) as growth promoters for swine. Benzoic acid and essential oils are increasingly be used for this purpose worldwide. The whole manuscript presents a brief and clear Introduction, Materials and Methods with the necessary information, quite interesting Results, a well-written Discussion that compare main results with other important articles in the field. Finally, the conclusion is direct pointing the main finding of the article.
However, I strongly suggest the authors to revise the whole manuscript to organize better some information, mainly in Materials and Methods and Results. In Material and Methods, all main topics are present, but the organization is not Ok. There are some paragraphs without a title (the first three paragraphs could be in a topic as for example Animals and Feed) as well as topics with only one or two sentences (as for example, pH of the gastrointestinal tract, volatile fatty acids and organ weight). The authors need to review all MM looking for a better organization. On oppose, there is no divisions in the Results to organize the information. I think the organization in topics would be necessary to improve the whole comprehension of the results.
Author Response
Dear Reviewer #3, we gratefully acknowledge your thorough evaluation and valuable comments. We do believe that it has contributed to improve the quality of our manuscript. All the changes in the manuscript made from your comments are highlighted in pink. Other general changes made by the authors, such as English or update results are standardized with the green color. In addition, changes proposed by more than one reviewer are marked by the color of the first reviewer who suggested such changes, indicating the lines where the changes were made.
The research question is important. It is necessary to investigate alternative additives to antibiotics (mainly for Colistin) as growth promoters for swine. Benzoic acid and essential oils are increasingly be used for this purpose worldwide. The whole manuscript presents a brief and clear Introduction, Materials and Methods with the necessary information, quite interesting Results, a well-written Discussion that compare main results with other important articles in the field. Finally, the conclusion is direct pointing the main finding of the article.
However, I strongly suggest the authors to revise the whole manuscript to organize better some information, mainly in Materials and Methods and Results. In Material and Methods, all main topics are present, but the organization is not Ok. There are some paragraphs without a title (the first three paragraphs could be in a topic as for example Animals and Feed) as well as topics with only one or two sentences (as for example, pH of the gastrointestinal tract, volatile fatty acids and organ weight). The authors need to review all MM looking for a better organization. On oppose, there is no divisions in the Results to organize the information. I think the organization in topics would be necessary to improve the whole comprehension of the results.
Thank you for your comments, we revised M&M and results section to improve the organization of the manuscript.

Reviewer 4 Report
Dear Authors
Please note that English is not my native language.
It is an unquestionable need to abandon the use of antibiotics as growth promoters. Although this argument is very explicit in the document, the need to replace the use of antibiotics by other products is not emphatically explained. That is, why do the authors consider necessary to replace colistin with a mixture of benzoic acid and EO's? Is animal health improved? Is the economic gain so relevant?
In my opinion, the papers should provide enough information to replicate the experiences. In this case, it is not possible to understand the exact composition of the EO's mixture. In "Materials and Methods" we realize that this is a mixture of various EO's provided by VeroVitall and VeroWin. What is the exact composition of each of these products? As a mixture of EO's, how much of each one is used in each treatment? On line 131 we are in doubt that it is only timol or a mixture of EO's.
Before slaughter, were the animals subjected to a period of fasting? In my opinion this period is important for the evaluation of digestive parameters such as VFA, pH, microbiota, organ weigh.
In my opinion, the collection of feces samples in the pre-killing phase and in the post killing should have been done in the same way, that is, in the post-killing phase, feces present in the most distal region of the large intestine should have been evaluated, for example, and not those from the cecum. This aspect may influence, for example, the composition of the microbiota.
In my opinion, the discussion should be improved. It should explain the results obtained logically and reasonably. In this paper the authors used bibliography that uses different parameters from their own to justify their data.
For exemple, in lines 327 and 328, if they did not evaluate digestibility, why do they justify their data with this parameter? What are the conditions that the authors consider that may have favored the BA+EO3 treatment? The conditions evaluated in this experiment, namely pH, VFA, microbiota, do not show significant differences between treatments. There are other conditions that justifiy this ?
Lines 332 to 339 – if the authors did not evaluate the effect of an isolated EO how to justify that the data obtained is explained by the joint action of essential oils? The same for line 352, if the authors did not evaluate digestibility how they justify the results with digestibility?
Lines 376 to 380 – this explanation extends to the effect of benzoic acid on the microbial population, reducing it, and this was not observed when we compared the NC and BA treatments, for example. Why?
Lines 52, 58, 68 and 85, the names of microorganisms must be in italics.
Author Response
Dear Reviewer #4, we gratefully acknowledge your thorough evaluation and valuable comments. We do believe that it have contributed to improve the quality of our manuscript. All the changes in the manuscript made from your comments are highlighted in grey. Other general changes made by the authors, such as English or update results are standardized with the green color. In addition, changes proposed by more than one reviewer are marked by the color of the first reviewer who suggested such changes, indicating the lines where the changes were made.
Dear Authors
Please note that English is not my native language.
It is an unquestionable need to abandon the use of antibiotics as growth promoters. Although this argument is very explicit in the document, the need to replace the use of antibiotics by other products is not emphatically explained. That is, why do the authors consider necessary to replace colistin with a mixture of benzoic acid and EO's? Is animal health improved? Is the economic gain so relevant?
Thank you for the comments, introduction and discussion were improved to explain the use of these additives combined.
In my opinion, the papers should provide enough information to replicate the experiences. In this case, it is not possible to understand the exact composition of the EO's mixture. In "Materials and Methods" we realize that this is a mixture of various EO's provided by VeroVitall and VeroWin. What is the exact composition of each of these products? As a mixture of EO's, how much of each one is used in each treatment?
We do not have the precise information about it, as we used a commercial product, we just evaluated the thymol content, I this sense we describe the information that we have from DSM that the products are a blend of essential oils includes thymol, 2-methoxyphenol, and eugenol with estimated total of 10%, and piperine and curcumin estimated total of 3%
On line 131 we are in doubt that it is only timol or a mixture of EO's.
We evaluated just the thymol composition. Bur other EO are present.
Before slaughter, were the animals subjected to a period of fasting? In my opinion this period is important for the evaluation of digestive parameters such as VFA, pH, microbiota, organ weigh.
The animals were not subject to a period of fasting.
In my opinion, the collection of feces samples in the pre-killing phase and in the post killing should have been done in the same way, that is, in the post-killing phase, feces present in the most distal region of the large intestine should have been evaluated, for example, and not those from the cecum. This aspect may influence, for example, the composition of the microbiota.
We agree, this will be considered in a next trial, collecting samples from the same regions.
In my opinion, the discussion should be improved. It should explain the results obtained logically and reasonably. In this paper the authors used bibliography that uses different parameters from their own to justify their data.
For exemple, in lines 327 and 328, if they did not evaluate digestibility, why do they justify their data with this parameter? What are the conditions that the authors consider that may have favored the BA+EO3 treatment? The conditions evaluated in this experiment, namely pH, VFA, microbiota, do not show significant differences between treatments. There are other conditions that justifiy this ?
BA e essential oils can improve digestibility and health parameters, this was better explained with other references in the discussion section.
Lines 332 to 339 – if the authors did not evaluate the effect of an isolated EO how to justify that the data obtained is explained by the joint action of essential oils? The same for line 352, if the authors did not evaluate digestibility how they justify the results with digestibility?
Other authors evaluated the use of this essential oils combined and determined better digestibility; in this sense we speculate that the use of the essential oils can improve digestibility.
Lines 376 to 380 – this explanation extends to the effect of benzoic acid on the microbial population, reducing it, and this was not observed when we compared the NC and BA treatments, for example. Why?
We did not find a plausible explanation for this finding. But It could be associated as the acid did not alter the composition but improvs the gut health.
Lines 52, 58, 68 and 85, the names of microorganisms must be in italics.
Thank you, it was corrected.

Reviewer 5 Report
In the manuscript " Benzoic acid combined with essential oils can be an alternative to the use of antibiotic growth promoters for piglets challenged with E. coli F4" authors are aimed at investigating the effect of combination of Benzoic acid and essential oils on growth performance and intestine health of weaned piglets challenged with E. coli F4. Based on reported findings, authors conclude that maternal combination of benzoic acid and essential oils supplementation at 3g/kg could offers improved performance, aside from being and economically viable alternative to replace Colistin. Although most of the reported results and discussion are sustained by data, some points need to be further clarified as reported in the following points below.
Comments for the authors:
- Line 92: why only barrows are used in this experiment?
- Line 105, 274: in table 1, crude protein level was 21.4% for pre-initial 1 period and 19.0% for pre-initial 2 period. This high protein level may lead to high diarrhea rate which was observed in table 3. And this may interfere with the effect of combination of benzoic acid and essential oils and also antibiotics supplementation.
- Line121-125: how does the author determine the supplementation dose of combination of benzoic acid and essential oils?
- Line 141, 175: in this experiment, piglets were challenged with E. coli F4. however, total Escherichia coli was evaluated which included not only E. coli F4 but also other strains. I hold that specific selective medium for E. coli F4 should be used or using quantitative PCR to detect abundance of E. coli F4.
- Line 149: what is the reason that author decide to slaughter at d 42?
- Line160-167: The diet switch occurred at d 31, but piglets were not weighted at this time point, why?

Author Response
Dear Reviewer #5, we gratefully acknowledge your thorough evaluation and valuable comments. We do believe that it have contributed to improve the quality of our manuscript. All the changes in the manuscript made from your comments are highlighted in red. Other general changes made by the authors, such as English or update results are standardized with the green color. In addition, changes proposed by more than one reviewer are marked by the color of the first reviewer who suggested such changes, indicating the lines where the changes were made.
In the manuscript " Benzoic acid combined with essential oils can be an alternative to the use of antibiotic growth promoters for piglets challenged with E. coli F4" authors are aimed at investigating the effect of combination of Benzoic acid and essential oils on growth performance and intestine health of weaned piglets challenged with E. coli F4. Based on reported findings, authors conclude that maternal combination of benzoic acid and essential oils supplementation at 3g/kg could offers improved performance, aside from being and economically viable alternative to replace Colistin. Although most of the reported results and discussion are sustained by data, some points need to be further clarified as reported in the following points below.
Comments for the authors:
Line 92: why only barrows are used in this experiment?
Only barrows were used to minimize the effect of the gender on the results, as it would be necessary to include more one factor on the statistical analysis
Line 105, 274: in table 1, crude protein level was 21.4% for pre-initial 1 period and 19.0% for pre-initial 2 period. This high protein level may lead to high diarrhea rate which was observed in table 3. And this may interfere with the effect of combination of benzoic acid and essential oils and also antibiotics supplementation.
We agree with this observation. The high level of protein could influence the results.
Line121-125: how does the author determine the supplementation dose of combination of benzoic acid and essential oils?
The supplementation levels were calculated based on the values that the DSM company says that is present on the product.
Line 141, 175: in this experiment, piglets were challenged with E. coli F4. however, total Escherichia coli was evaluated which included not only E. coli F4 but also other strains. I hold that specific selective medium for E. coli F4 should be used or using quantitative PCR to detect abundance of E. coli F4.
We agree that a more specific evaluation of the E. Coli present should be performed, in future trials it would be considered.
Line 149: what is the reason that author decide to slaughter at d 42?
This was defined to evaluate if the supplementation of the additives would be beneficial in recovery the TGI of the piglets after the challenge.
Line160-167: The diet switch occurred at d 31, but piglets were not weighted at this time point, why?
As was established that the performance should to be evaluated just on days 35, 42, 49, and 63 this was not considered to weight the animals at the feed transition.

Round 2
Reviewer 2 Report
The Authors considered most of the Reviewer's comments but English quality still requires improvement before publication. The Introduction lacks information on a dose of essential oils used in other studies, although the Authors claim that is was added. In line 63 "of colistin" is repeated. PLease, delete one. "Equal" is not a good word in description of results because there are differences in numbers. Please replace it with other words or rewrite the description of results. It is recommended to indicate that there were no differences between groups. The first sentence of the Discussion is too long and should be divided in two.
Author Response
Dear Reviewer #1, we gratefully acknowledge your thorough evaluation and valuable comments. We do believe that it has contributed to improve the quality of our manuscript. All the changes in the manuscript made from your comments are highlighted in blue and marked with track changes. Other general changes made by the authors are standardized with the green color and marked with track changes. In addition, changes proposed by more than one reviewer are marked by the color of the first reviewer who suggested such changes, indicating the lines where the changes were made.
Comments to author:
The Authors considered most of the Reviewer's comments but English quality still requires improvement before publication. The Introduction lacks information on a dose of essential oils used in other studies, although the Authors claim that is was added. In line 63 "of colistin" is repeated. PLease, delete one. "Equal" is not a good word in description of results because there are differences in numbers. Please replace it with other words or rewrite the description of results. It is recommended to indicate that there were no differences between groups. The first sentence of the Discussion is too long and should be divided in two.
Thank you for the comments.
- English was reviewed by an expert.
- The word colistin was removed line 65.
- Word equal was changed by similar line 286.
- The first sentence of the discussion was corrected.
Reviewer 4 Report
Dear Editor
Most of my doubts have not been adequately clarified.
In my opinion, the experimental design is not appropriate. I consider that some procedures must be standardized: the animals had to be fasted in order to properly evaluate the digestive parameters; the collection of stool samples should be carried out in the same way in the different phases (pre-Killing phase and post-killing phase).
In my opinion the discussion remains very speculative.
It must be reviewed by an english native speaker.
Author Response
Dear Reviewer #2, we gratefully acknowledge your thorough evaluation and valuable comments. We do believe that it has contributed to improve the quality of our manuscript. All the changes in the manuscript made from your comments are highlighted in pink and marked with track changes. Other general changes made by the authors, such as English or update results are standardized with the green color and marked with track changes. In addition, changes proposed by more than one reviewer are marked by the color of the first reviewer who suggested such changes, indicating the lines where the changes were made.
Comments to author:
Most of my doubts have not been adequately clarified.
Sorry for this, we tried to improve the manuscript using your comments below.
In my opinion, the experimental design is not appropriate. I consider that some procedures must be standardized: the animals had to be fasted in order to properly evaluate the digestive parameters; the collection of stool samples should be carried out in the same way in the different phases (pre-Killing phase and post-killing phase).
Unfortunately we cannot change the experimental design now, thank you for your consideration, but considering that the animals should be fasted we do not agree, as we want to evaluate the effect of the diet on the digestive parameters and a fasting period could impact on it. Regarding the samples we do agree it should have been made in the same way, it would be considered for future studies.
In my opinion the discussion remains very speculative.
Thank you for the comment. All the discussion was based on similar trials, in this sense, we tried to elucidate the mechanisms and understand how the effects occurred.
It must be reviewed by an english native speaker.
The English was reviewed by an expert.